

# Lake outbursts of the eastern part of the Larsemann Hills, East Antarctica, through snow and ice dams

Alina Boronina[1], Sergey Popov[2,1], and Galina Pryakhina[1]

[1]Saint-Petersburg State University, 7-9 Universitetskaya Emb., St. Petersburg 199034, Russia
[2]Polar Marine Geosurvey Expedition, 24 Pobedy str., St. Petersburg-Lomonosov 198412, Russia

**Correspondence:** A. Boronina (al.b.s@yandex.ru)

**Abstract.** The Antarctic oasis Larsemann Hills is characterized by a developed drainage system. It includes several lakes of different genesis. However, besides the state of knowledge of this region and regular expeditions, which are conducted nowadays, the lakes of this oasis have not been studied comprehensively yet. In general, international and Russian research is dedicated to the monitoring of the ecological state of the water bodies, inferring climate change signals from the lake deposits.

At the same time, works related to determination of the bathymetry and morphology of water bodies do not virtually exist or are hidden in scientific-technical reports of the Antarctic Programs of different countries. Interest in the investigation of the oasis lakes has increased sharply after the formation of a vast depression on on Dålk Glacier (Larsemann Hills, East Antarctica) on 30 January 2017 caused by the outburst of intraglacial reservoir. Field research in 2017/18 revealed that sudden destructions of impound dams and the generation of breakthrough floods are indicative for many lakes of the oasis. Thus, the present work

aims at the application of mathematical modeling methods to shed light on the processes that lead to dam destruction and the outburst of lakes temporarily impounded by natural firn-ice and glacial dams. Such discharges are comparable to *jökulhlaups* and can be calculated using the adapted model of *Yu.B. Vinogradov*. As main objects of this research we select among the lakes located close to Russian and foreign Antarctic stations and field bases those, for which destructions of ice-snow bridges are detected. According to the modelling results, the following characteristics were identified for every outburst: the distribution of

the discharges over time, the volume and transmission time of the flood. Moreover, its catastrophic risk and fracture force was assessed. The data obtained will form a basis for studying the formation of temporary ice hydrographic networks in Antarctica and the analogous processes occurring under the Arctic glaciers.

## 1   Introduction

The characteristic feature of the eastern part of the Lasermann Hills (East Antarctica) is the abundance of lakes, which can be explained by the presence of a young structural exaration relief and an undeveloped drainage hydrographic system. A part of these water bodies formed as a result of the ponding of tectonic valley depressions and by snowfields as well (Simonov, 1971).




Since the lakes are mainly fed by snow melt, the most intensive influx of water happens during the short Antarctic summer. During this period of time a significant increase in water level and a destruction of snow and ice dams, followed by the generation of outburst floods can be observed (Popov et al., 2017, 2018; Boronina et al., 2018, 2019, in press). Affecting severely the local infrastructure as well as transport and logistic operations, such phenomena are recognized as especially dangerous and, therefore, relevant to study. Different methods, such as field studies, airborne photography interpretation, physical modelling (Tingsanchali and Chinnarasri, 2001; Zhu et al., 2004, 2011; Mohamed and El-Ghorab, 2016; de Haas et al., 2018; Wang et al., 2018) and mathematical modelling (Röthlisberger, 1972; Nye, 1976; Björnsson, 1992, 1998; Walder and Fowler, 1994; Clarke, 2003; Fowler, 2009; Hewitt, 2011), are used for the research of outburst floods.

Fieldworks *in situ* allow to gather significant information about a catastrophic phenomenon which has already occurred repeatedly in the past and can be anticipated to reoccur in the future. In particular, after an outburst of a glacial lake it is possible to determine the boundaries of the disaster zone, the pathway and means of flux of the outflow stream, and to estimate parameters describing the outburst. However, many reservoirs are difficult to reach for the implementation of fieldwork. Physical modelling is employed in those cases when it is necessary to determine quantitative characteristics of the hazardous phenomenon but it is impossible to perform full-scale field measurements. However, frequently this method is also complicated to realize because of the difficulties of experimental constructions. In the case of mathematical modelling there is no need in building expensive installations and the researchers have the opportunity to define a wide spectrum of the quantitative characteristics. That is why the named method found wide application in scientific studies. We develop a mathematical model to derive hydrographs for the scenario of abrupt water outbursts from the lakes in the eastern part of the Larsemann Hills. This region is of particular interest from a practical point of view. An extensive infrastructure including the Russian Antarctic Progress Station, an airfield and the logistic base of the traverse to the inland of Antarctica is situated there. In that area an extensive system of reservoirs is found on the Broknes Peninsula (Fig. 1) which cover not only bedrock but also the glacier and extent beneath it as well (Simonov, 1971; Gillieson et al., 1990; Gasparon and Matschullat, 2006; Hodgson et al., 2006; Popov et al., 2018). As the objects of the presented research we have chosen those lakes, on which destructions of snow and ice dams are recorded according to unpublished scientific technical reports of the Russian Antarctic Expedition (RAE). One example is the relatively small Lake Discussion in the central part of the Broknes Peninsula near the western coast of Nella Fjord. Analyzing recent cartographic data (Antarctic Xiehe Peninsula, 2006; Larsemann Hills, 2015) and also during the reconnaissance survey in the field season of 63[rd] RAE (2017/18) the authors have discovered evidence for frequent outbursts. The inflow into the reservoir follows the stream that originates in Lake LH59 and receives the meltwater of the snowfields located in the water-collecting territory of the investigated reservoir. The prechannel flow from the lake is highly variable, probably this is the reason for the frequent overflow of its bedrock basin.

According to earlier observations *(not published)*, a similar situation can also be seen at the lake LH73. The reservoir is located in an elliptic pan in close proximity to the lake system Progress and Sibthorpe. In the case that the hydraulic system of the three mentioned reservoirs bursts out, the results can have a catastrophic character. Such an event was registered in 2004 when the water flow destroyed part of a transport route across the snowfield near the lakes that connects Progress Station with the airfield. The most recent outburst of the Lake LH73 happened in March 2017. Due to lack of monitoring the variation value





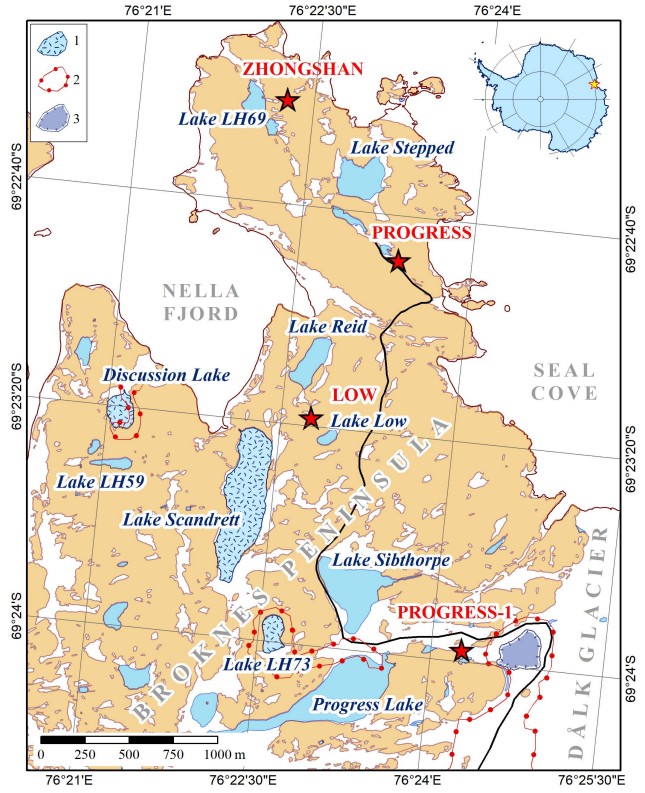

**Figure 1.** Location chart of the hydrological works in the eastern part of Broknes Peninsula. 1 – bathymetric surveys; 2 – tachymetric surveys; 3 – depression on Dålk Glacier. Australian Antarctic Division GIS dataset has been used for the map preparation.

of the water level is unknown. However, during our reconnaissance, cracks that mark the waterline before the outburst, were clearly visible on the ice near the coastline. Furthermore, in January 2018 the authors recorded a rapid drop in water level of

Lake Scandrett as a result of the destruction of a snowfield in the northeastern part of the reservoir.

## 2   Data acquisition and processing

During the field works of the $63^{rd}$ RAE a complex of hydrological and geodetic measurements was completed. On the investigated lakes bathymetric survey and observations of the water level were performed. Some coastal areas above the high water level was mapped by tachymetric surveys as well. The water depths were measured with the electronic acoustic sounder

Garmin GPSMAP 585 *(Garmin Ltd., Taiwan)*. In addition, the shoreline of the lakes was traced with a satellite receiver Garmin GPSmap64st *(Garmin Ltd., Taiwan)*. The tachymetric survey of coastal areas was performed using the tachometer Trimble M3 DR 5" *(Trimble Navigation, Ltd., USA)*.

Data on the bottom relief and changes of the water level during the outburst allowed to make bathymetric models of the water bodies before the outburst and estimate their morphometric characteristics. The cartographic program Surfer 15 *(Golden*





*Software Inc., USA)* with the *Kriging* algorithm has been used for gridding the data. A bathymetry grid is derived for Discussion
Lake based on 2762 measurements. The average error of the grid is 0.015 m (without account of systematic errors). The total
interval of the errors is from –0.785 m to 0.709 m. The standard error was estimated at 0.12 m. The bathymetry grid for Lake
LH73 has been derived from 260 points. The average error was –0.03 m with variations from –1.27 m to 0.42 m. The standard
error was estimated at 0.19 m. The same grid for Lake Scandrett was created on 38 points. The average error was 0.052 m with
the error interval between –0.967 m and 1.15 m with a standard error of 0.55 m. We conclude, therefore, an adequate quality
of the grids (and charts).

## 3 Modelling of the lake outbursts

Currently, there are a large number of mathematical models describing the formation and development of floods (Nye, 1976;
Björnsson, 1992, 1998; Walder and Fowler, 1994; Clarke, 2003; Flowers et al., 2004; Evatt et al., 2006; Evatt and Fowler,
2007; Fowler, 2009; Hewitt, 2011). However, when focusing on Polar Regions, the majority of them describe processes that
occur under the ice cover or inside it. For instance, *H. Röthlisberger* (1972), *J. Nye* (1976) and *R.J. Thayyen* (2011) consider the
water flow in channels and tunnels inside a glacier. On the other hand, the models suggested by *H. Björnsson* (1992; 1998) and
*A. Fowler* (2009) are well suited for describing the outburst of Grímsvötn lake located under the ice cap Vatnajökull in Iceland.
The applicability of calculation methods for modelling of floods during the outbursts of subglacial reservoirs of Antarctica
was dealt with in (Marchant et al., 2011; Pattyn, 2013). Among previously described models we chose the one suggested
by *Yu.B. Vinogradov*, which is suitable for outbursts of lakes located at the glacier surface, temporarily impounded by snow
and ice dams (Vinogradov, 1976). Its main advantage is that input information and parameters can be gathered during field
works without the need of the empirical relation calculations. For this reason, this particular model was taken as a basis for
modelling the outbursts of the lakes of the eastern part of the Larsemann Hills. It was adopted for the conditions in Antarctica
by taking into account the possible presence of ice cover over the reservoir. This makes it fundamentally suitable for studying
the processes occurring in the case of hypothetical outbursts of subglacial reservoirs and intraglacial cavities filled with water
masses.

  The mathematical model is based on the numerical solution of two principle equations: *the continuity equation* and *the
energy conservation law*. The detailed description of the model can be find in (Popov et al., 2019, in press). The initial data
for the modelling are: a bathymetric model of the lake before the outburst, synthesized into a grid; the difference in elevation
between the points of entry and exit from the tunnel; the length of the tunnel; the temperature of the water in the lake before the
outburst; the thickness of the glacier above the lake, as well as the density and specific heat of the dam material. The constants
for the simulation are: specific heat capacity of water $c_w = 4190$ J kg$^{-1}$ °C$^{-1}$, its density $\rho_w = 1000$ kg m$^{-3}$, specific heat
of ice melting $\lambda = 3.34 \times 10^5$ J kg$^{-1}$. The density of the icy dam through which the outburst occurs is $\rho_i = 910$ kg m$^{-3}$.
In all the simulated scenarios, the parameter was determined by calculation, based on the length of the tunnel (Vinogradov,
1976; Popov et al., 2019, in press), and the dependence of the volume of the lake on the depth was calculated from the
bathymetric grid. Results of the modelling is an outburst flood hydrograph, at the exit of a snowfield or glacier, through which





a sudden discharge of lake waters occurs. According to the modelling results for each outburst, the following characteristics are determined: fluctuations of water discharge with time, the volume of the flood and the time of its passage.

## 3.1 Discussion Lake area

Discussion Lake is located in an almost circular pan. The coastline is not much indented (Fig. 2). The length of this lake before the outburst in January 2018 was 230 m, with a maximum width of about 150 m. The water surface area was estimated at $24,540 \mathrm{~m}^2$, with a water volume of $601,440 \mathrm{~m}^3$. There are snowfields of considerable size to the north and west of the lake. The latter constitutes the dam for the lake water. The maximum depth of the reservoir, before the outburst of lake water was 4.8 m, with an average depth of about 2.4 m. The area of greatest depths is located in the central part and has a circular shape.

The outburst of the Discussion Lake on 22 January 2018 took place along a tunnel formed in a snow-ice dam that pounded the reservoir. The dimensions of the tunnel vary widely. Its maximum height was estimated at 3 m, with a width of more than 8 m. By Antarctic mid-summer the reservoir was almost completely free of ice. According to the field measurements, the length of the tunnel, through which the outburst occurred, is 128 m. The height difference between the entrance and exit of the tunnel is 2.17 m. The water temperature was assumed to be $5°\mathrm{C}$ based on the field observations.

Results of the modelling are depicted in Figure 3. The hydrograph is characterized by a smooth shape, without abrupt changes in water flow. The maximum flow rate is estimated at $1.8 \mathrm{~m}^3 \mathrm{~s}^{-1}$ and is reached 7 hours and 37 minutes after the start of the flow. The volume of water that constituted the flood is $26,750 \mathrm{~m}^3$. The total flow time into the Nella Fjord is 9 hours and 40 minutes.

During the reconnaissance survey, it was found that the outflow of lake water along the derived path occurs quite frequently. Clearly visible high water marks from outflows of varying intensity are preserved along the tunnel walls. This is evidenced by the stepped structure inside the snowfield (Fig. 4). Thus, it is concluded that the maximum dimensions of the tunnel were formed by the flow of previous years, and that the most recent outflow was not disastrous. This assumption is well confirmed by the results of the modelling.

## 3.2 Lake LH73 area

Lake LH73 is located in an oval pan, in close proximity to the lakes Progress and Sibthorpe. Its length before the outburst in March 2017 was 250 m, with a maximum width of about 160 m (Fig. 5). The water surface area of the reservoir was estimated at $31,640 \mathrm{~m}^2$, with a water volume of about $84,000 \mathrm{~m}^3$. The areas of greatest depths are located in the central part of the lake and have an elongated shape that extends from north to south. The maximum depth before the discharge of lake water was 4.9 m. The flanks of the valley of the reservoir as a whole are flat, of small slope and poorly dissected. Along the southern shore the lake verges on a snowfield of considerable size. The snow-ice wall in this place is almost steeply. Periodically, in the warm season of the year, blocks of ice and snow break off from it, forming relatively large caverns and depressions.

The length of the tunnel through which the outburst occurred in March 2017 was approximately estimated at 480 m. The height difference between the entry and exit tunnel points was 27.9 m. The water temperature in the lake at the time of destruction of the dam is unknown. However, guided by the fact that the described events took place in March during the

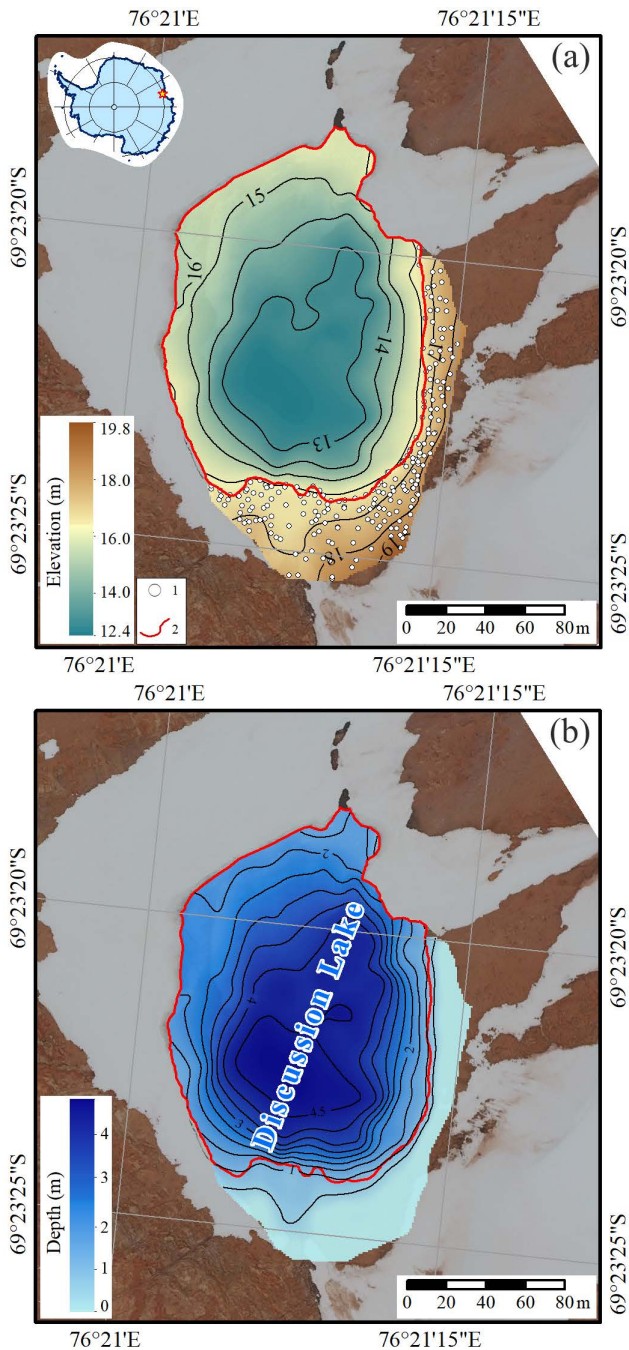

**Figure 2.** Bedrock topography **(a)** and the depth of Discussion Lake before the outburst **(b)**. 1 – points of the height measurements; 2 – coastline of the lake after the outburst. Airborne photomap compiled by *A. Mirakin* (RAE) has been used.

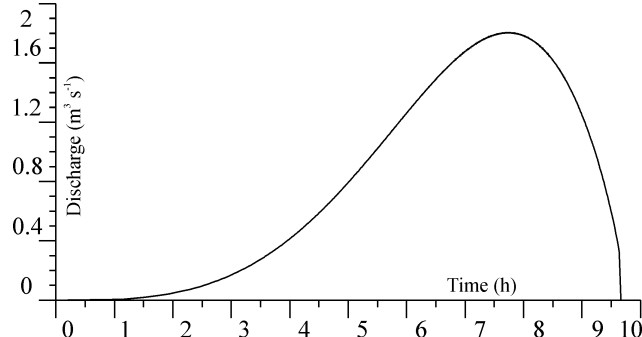

**Figure 3.** Hydrograph of the flood from Discussion Lake.

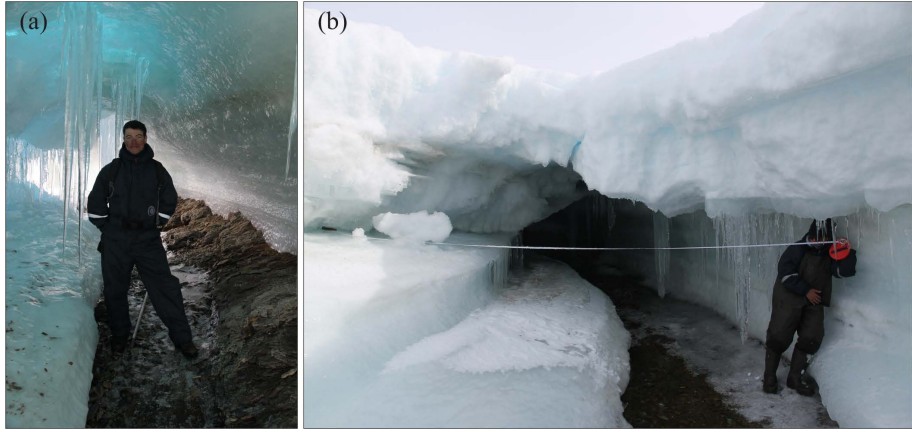

**Figure 4.** The outburst tunnel at Discussion Lake. Photos by *S. Popov* and *A. Boronina*, January 2018.

freezing season, it was taken close to zero. The ice thickness above the lake is zero since there was no ice cover. The water volume that formed the flood from Lake LH73 was $46,800$ m$^3$. At the same time, the water level decreased by about 1.6 m. The calculated hydrograph is depicted in Figure 6.

The hydrograph is asymmetric. A gentle rise ends with a rather sharp decline and cessation of outflow. The highest water
discharge reaches a value of $0.71$ m$^3$ s$^{-1}$. The outflow of the specified volume of water lasted about 4 days. Since there are no observations on the lake during winter season, it is very difficult to compare the calculated hydrograph with the real one. However, according to *A. Teplyakov (2018, private communication)*, the water that broke through flowed from Lake LH73 to Progress Lake and froze onto the ice surface (Fig. 7a). Based on this, the estimated water flows seem to be close to reality and the outflow took place in the form of a small and gentle stream.

The outburst calculations for Lake LH73 are of particular interest because, as mentioned earlier, the lakes Progress, LH73 and Sibthorpe make up a joint hydrological system and the outburst of the first reservoir one is usually catastrophic. RAE informs about a similar, abrupt discharge of Progress Lake in late February 2018, after the conclusion of the summer field



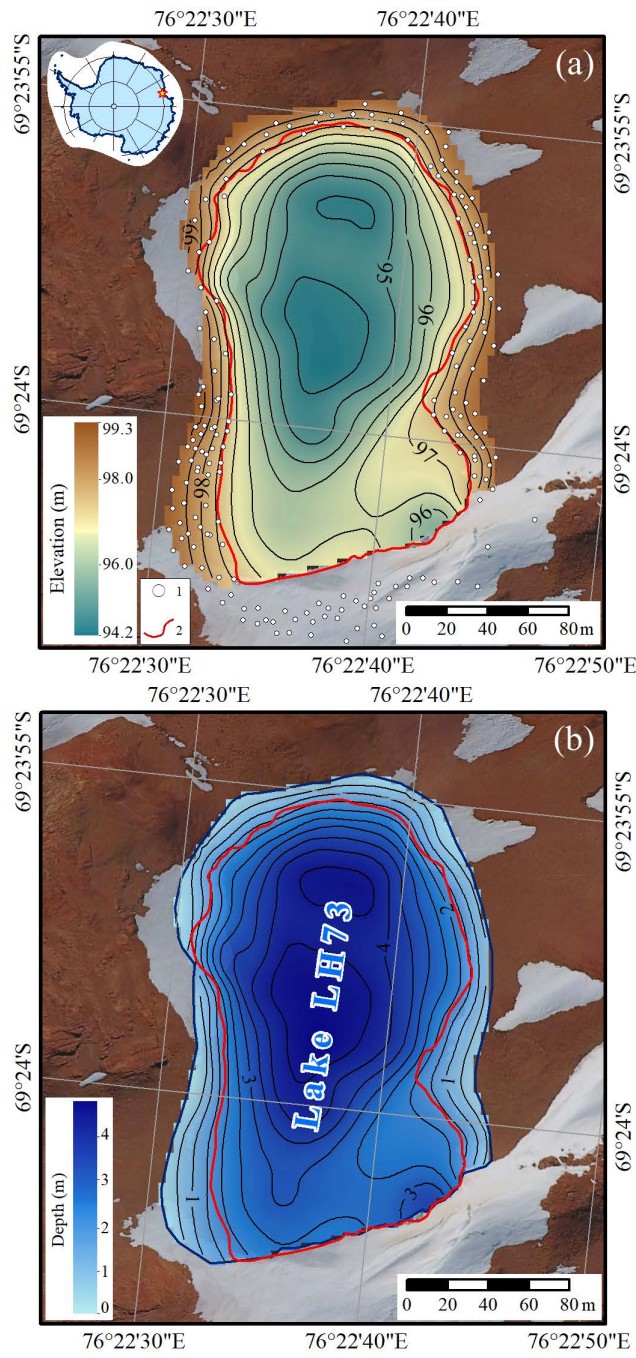

**Figure 5.** Bedrock topography **(a)** and the bathymetry of Lake LH73 before the outburst **(b)**. 1 – points of the height measurements; 2 – coastline of the lake after the outburst. Airborne photomap compiled by *A. Mirakin* (RAE) has been used.

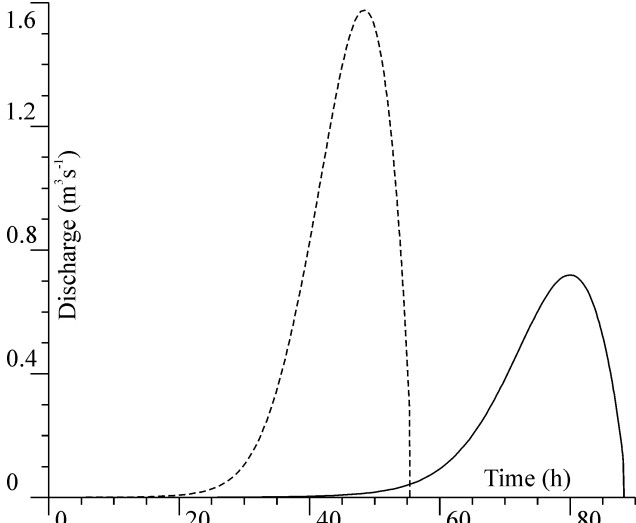

**Figure 6.** Hydrograph of the flood from Lake LH73.

season. It resulted in the destruction of a snow dam which connects the route from the field base Progress-1 to the airfield (Fig. 7b). The water flow reached a width of 4 m and a depth of 1.5 m.

Based on this evidence, the scenario of an outburst of the total water volume contained in the basin of Lake LH73 occurs was modeled. The hydrograph is presented in Fig. 6. The highest water flux in this case is $1.6 \text{ m}^3 \text{ s}^{-1}$ which is reached two days after the start of the flow. The volume of the outburst flood was estimated at $84,000 \text{ m}^3$. The resulting overflow of water, in turn, would cause a sharp increase in the water level in the Lake Progress and an accelerated destruction of the snow-ice dam. In addition, it is necessary to understand that these values are obtained for the case of an outburst during the autumn season.

If the destruction of the dam occurs during the Antarctic summer, the flow capacity will be much stronger and the resulting depressions at the snow surface may interrupt the logistics of the station and the airfield for a long time.

### 3.3   Lake Scandrett area

Lake Scandrett is one of the largest reservoirs of the peninsula. It is fed by water coming directly from the glacier along a small stream about 1.5 km long. Due to the considerable depths, the complete ice clearance of the reservoir does not occur even

during the Antarctic summer. Usually, by the beginning of January the ice covers about 80% of the lake surface. According to the survey, the length of the Lake Scandrett is 891 m with an average width of about 180 m. The water-table area is estimated at $157,920 \text{ m}^2$, and the water volume before the outburst in December 2017 reached $1,544,370 \text{ m}^3$. The lake bottom is composed of two basins. The larger one is located at the lake center and has a circular shape (Fig. 8). The maximum depth before the outburst was 17.9 m. The smaller basin is located in the southwestern part of the lake and adjoins the coast. The western slopes

are steep, therefore large depths predominate even close to the shore. The eastern part of the lake is shallower, however, it is characterized by sharper drops in bottom levels and by significant slopes.



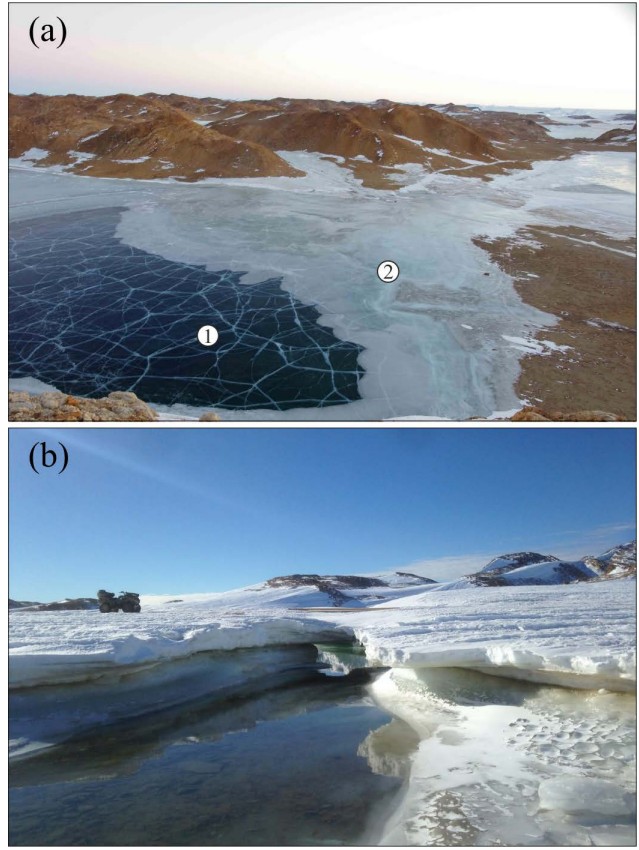

**Figure 7.** Implications of Progress Lake outburst. Section **(a)**: 1- ice on the lake, which was formed during the winter period; 2- water frozen on the ice surface as a result of an outburst. Section **(b)**: depressions in the snowfield, formed by the outburst of Progress Lake in February 2018. Photos by: (a) *A. Teplyakov* in March 2017, (b) *A. Konyaev* in February 2018.

The length of the tunnel, through which the water escaped Lake Scandrett on 31 December 2017, was estimated to be about 130 m. The elevation difference, according to barometric leveling, was 16.6 m. At the moment of the outburst, the lake was practically completely covered with ice and the water temperature was taken close to $0°C$. According to the field observations, the drop in lake level of Lake Scandrett lasted for 3 days, which is in good agreement with the result of the modelling. The hydrograph is shown in Figure 9. The flood volume was estimated at $53,700 \mathrm{~m}^3$. The calculated total discharge time is 77 hours and 30 minutes. The maximum water flow amounts to $0.55 \mathrm{~m}^3 \mathrm{~s}^{-1}$ and is reached 58 hours after the onset of the flood. The outburst stream was sufficiently intense to produce a tunnel in the snowfield, which caused a subsidence of the snow surface. All the water flowed into the Nella Fjord.




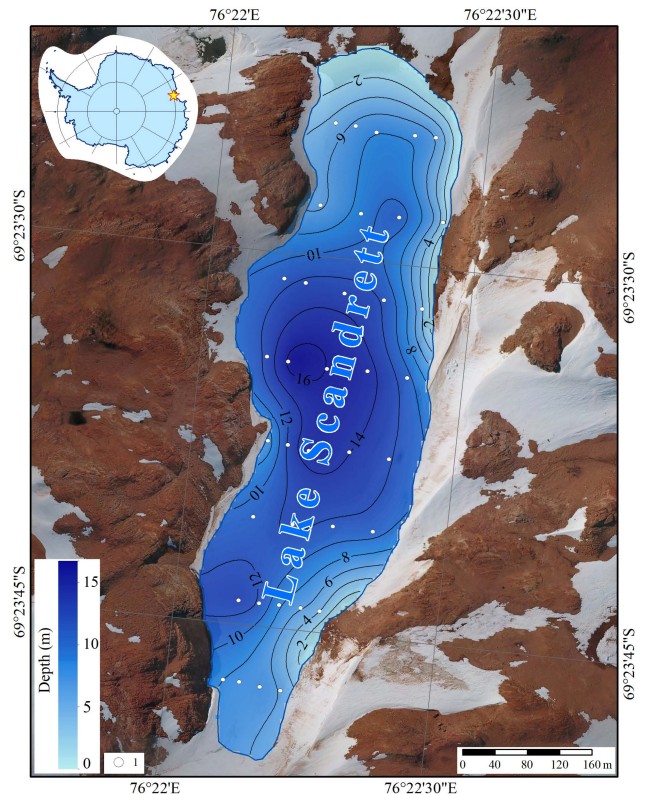

**Figure 8.** The bathymetry of Lake Scandrett before the outburst. 1 – points of the depth measurements. Airborne photomap compiled by *A. Mirakin* (RAE) has been used.

## 4 Conclusions

The results of our modelling reveal, for the first time, a major feature of the hydrological regime of the water bodies of the Antarctic oases: the discharge of lake water during the spring-summer period through tunnels in snow-ice dams. Depending on the morphology and morphometric characteristics of the lake basin, as well as the seasonal water level cycle, the outbursts are characterized by a varying duration (from several hours to several days) and recurrence regime (from annually repeated to non-periodic). At the moment, it is obvious that the trigger of such processes is the magnitude of the influx of melt water into the lake basin. In the context of a warming climate, this phenomenon will not only repeat, but is likely to become more widespread and to increase in magnitude. Under these conditions, the use of mathematical modeling methods allows to forecast consequences for a variety of scenarios.

The outburst hydrographs modeled here do not contradict the physical essence of the process: a gradual increase values of discharge during the rupture of the dam, the formation of a drainage tunnel and a rapid decline after reaching the maximum water flow. This suggests that the technique used in this work can be applied not only to the lakes along the Antarctic coast, but also to lakes of similar genesis in other glaciated environments in the Arctic and high mountains.

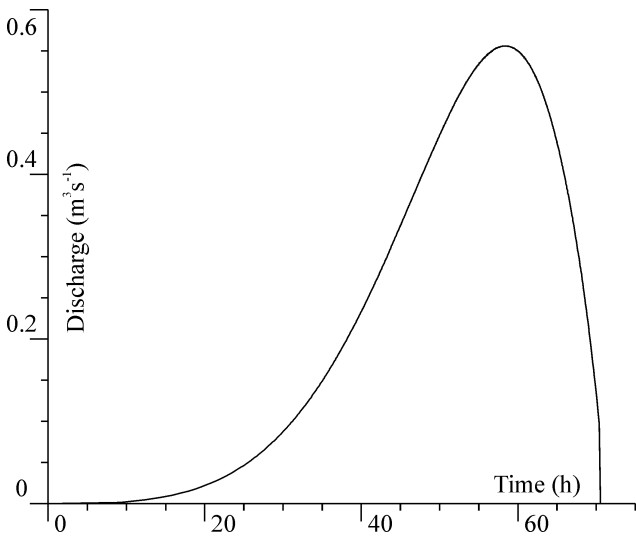

**Figure 9.** Hydrograph of the flood from Lake Scandrett.

The combination of the results of both field surveys and mathematical modeling also demonstrate for the first time quantitatively the principal characteristics of the outburst floods: water volumes, discharge duration, variation in discharge flux with time. In addition to the scientific interest, these data are of practical importance in terms of life safety of Russian and foreign wintering stations and field bases.

*Data availability.* on request

*Author contributions.* *A. Boronina*: collecting the data during the fieldwork, processing the data, maps creation; *S. Popov*: the fieldwork organization, collecting the data during the fieldwork, processing the data; *G. Pryakhina*: common curation of the scientific work.

*Competing interests.* No competing interests are present

*Acknowledgements.* The Authors thank their colleagues *S. Grigorieva*, *G. Deshevykh*, *A. Sukhanova* for their valuable support, and also the chief of the Progress Station 63[rd] RAE *A. Voevodin* for his assistance in the realization of the field work; the chief of the station Progress 62[nd] RAE *A. Mirakin* for the photo and video materials; the staff of the Institute of Earth Sciences of St. Petersburg State University, Department of Hydrology for the provided hydrometric equipment; Australian Antarctic Division Data Center for GIS data for the Larsemann Hills area.



We also are grateful for the contributions of *A. Richter* which helped to improve and to clarify the manuscript. This scientific work was supported by the Russian Foundation for Basic Research in the framework of the scientific project No 18-05-00421.



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
