# Peer review of "Lake outbursts of the eastern part of the Larsemann Hills, East Antarctica, through snow and ice dams"

_The Cryosphere, 2019_

## Referee Comment (RC1) · Anonymous Referee #1 · 6 Oct 2019

The manuscript presents the novel interesting data on abrupt water outbursts observed in the eastern part of the Lasermann Hills (East Antarctica) related with lakes blocked by snow/ice dams. It fully confirms the scope of TC journal and has scientific and practical value for understanding GLOF phenomena at the Antarctic Ice Sheet marginal zone and protecting the research stations in the study area. The authors state (lines 9-11) that their aim is to apply "...mathematical modeling methods to shed light on the processes that lead to dam destruction and the outburst of lakes temporarily impounded by natural firn-ice and glacial dams". The model applied allows to estimate flood hydrographs using as input lake bathymetry, water temperature, tunnel length and its entry and exit points elevation. Regretfully, details of the model used are not

given but only a short description with reference to the authors paper published in Russian, that is not enough to allow the reproduction of results. As there are no data on measured hydrographs of outbursts, it is difficult to assess how well the model hydrographs reproduce the reality and how they might shed the light on the processes that lead to dam destruction. The abstract is too general and does not show the major specific findings including key quantitative results, but only states (lines 14-16) that," ... the following characteristics were identified for every outburst: the distribution of the discharges over time, the volume and transmission time of the flood. Moreover, its catastrophic risk and fracture force was assessed". The same is true concerning the Conclusions. The overall presentation is not well structured and clear. The data of observations and measurements are mixed with modelling results in one Section 3 Modelling of the Lake Outbursts. I suggest to give all observational and measurement data in one separate section for all three lakes, and then the modelling section with hydrographs (in one Figure with a,b,c panels) and a summary table with major results of modelling (outburst volumes, duration, maximum discharge). In large, now the manuscript is a kind of raw scientific correspondence with data on bathymetry of lakes and some information on outbursts, rather than a complete scientific article. The manuscript requires a thorough English language and organizational editing.

19-20: the sentence "A part of these water bodies formed as a result of the ponding of tectonic valley depressions and by snowfields as well" is not clear. 24-23: what is amount of snow melting (measured or estimated) in the study area? 54, 106, 126: Meaning of the term "pan" is depression/basin? 94: (Popov et al., 2019, in press) is already published. 115: when and where the water temperature was measured? This type of measurements is not mentioned in the Section 2 Data Acquisition and Processing. 147: Caption of Fig. 6 does not explain two different curves (dashed and solid).

---

## Author Comment (AC1) · 10 Dec 2019

Dear Reviewer,

The authors are grateful for the careful reading of the research article and constructive comments. We agree that they will make the manuscript better. We are ready to make all necessary changes after we receive answers from the remaining reviewers.

Kind regards, Alina Boronina.
* * *

---

## Referee Comment (RC2) · Anonymous Referee #2 · 21 Feb 2020

This manuscript describes observations and analysis of lake drainage events of periglacial lakes in East Antarctica. While the topic is highly interesting due to several reasons (e.g. relation to climatic variations, effect on the local environment), the manuscript severely lacks in clarity, scientific confirmability and valid conclusions.

The structure of the manuscript is rather unclear with information about the lakes, their setting, geometry and environmental conditions spread across different sections. Data are not clearly presented (e.g. no information about the bathymetry tracks across the lakes), figures miss important information (e.g. locations of the drainage channels) and the presentation of the results is spread over the text without a concise summary

(structured tables would probably help a lot to show the main findings instead of long sentences filled with numbers).

The modelling approach is not presented at all. Instead there is only a reference to the relevant paper. There needs to be at least a short presentation of the physical basis, the assumptions taken and the restrictions of the model. Otherwise, the results cannot be evaluated with respect to their reliability.

Also, relevant literature to ice dam failures and drainage are not found in the manuscript (see some examples below). Even though the observations are rather valuable and there is a large potential for obtaining more insights into such type of failures, the conclusions are rather weak and do not exploit the wealth of information which could be deducted from a thorough analysis.

Raymond, C. F., & Nolan, M. A. T. T. (2000). Drainage of a glacial lake through an ice spillway. IAHS publication, 199-210.

Carrivick, J. L., Tweed, F. S., Ng, F., Quincey, D. J., Mallalieu, J., Ingeman-Nielsen, T., ... & Russell, A. J. (2017). Ice-dammed lake drainage evolution at Russell Glacier, West Greenland. Frontiers in Earth Science, 5, 100.

Kingslake, J., Ng, F., & Sole, A. (2015). Modelling channelized surface drainage of supraglacial lakes. Journal of Glaciology, 61(225), 185-199.

Mayer, C., & Schuler, T. V. (2005). Breaching of an ice dam at Qorlortossup tasia, south Greenland. Annals of Glaciology, 42, 297-302.